# Improvement of Properties and Performances of Polyethersulfone Ultrafiltration Membrane by Blending with Bio-Based Dragonbloodin Resin

**DOI:** 10.3390/polym13244436

**Published:** 2021-12-17

**Authors:** Aulia Chintia Ambarita, Sri Mulyati, Nasrul Arahman, Muhammad Roil Bilad, Norazanita Shamsuddin, Noor Maizura Ismail

**Affiliations:** 1Doctoral Program, School of Engineering Science, Syiah Kuala University, Banda Aceh 23111, Indonesia; auliachinta@gmail.com; 2Department of Chemical Engineering, Syiah Kuala University, Banda Aceh 23111, Indonesia; nasrular@unsyiah.ac.id; 3Faculty of Integrated Technologies, Universiti Brunei Darussalam, Bandar Seri Begawan BE1410, Brunei; roil.bilad@ubd.edu.bn (M.R.B.); norazanita.shamsudin@ubd.edu.bn (N.S.); 4Faculty of Engineering, Universiti Malaysia Sabah, Kota Kinabalu 88400, Sabah, Malaysia

**Keywords:** antifouling, bio-based additive, dragonbloodin, polyethersulfone, ultrafiltration

## Abstract

Polyethersulfone (PES) is the most commonly used polymer for membrane ultrafiltration because of its superior properties. However, it is hydrophobic, as such susceptible to fouling and low permeation rate. This study proposes a novel bio-based additive of dragonbloodin resin (DBR) for improving the properties and performance of PES-based membranes. Four flat sheet membranes were prepared by varying the concentration of DBR (0–3%) in the dope solutions using the phase inversion method. After fabrication, the membranes were thoroughly characterized and were tested for filtration of humic acid solution to investigate the effect of DBR loading. Results showed that the hydrophilicity, porosity, and water uptake increased along with the DBR loadings. The presence of DBR in the dope solution fastened the phase inversion, leading to a more porous microstructure, resulted in membranes with higher number and larger pore sizes. Those properties led to more superior hydraulic performances. The PES membranes loaded with DBR reached a clean water flux of 246.79 L/(m^2^·h), 25-folds higher than the pristine PES membrane at a loading of 3%. The flux of humic acid solution reached 154.5 ± 6.6 L/(m^2^·h), 30-folds higher than the pristine PES membrane with a slight decrease in rejection (71% vs. 60%). Moreover, DBR loaded membranes (2% and 3%) showed an almost complete flux recovery ratio over five cleaning cycles, demonstrating their excellent antifouling property. The hydraulic performance could possibly be enhanced by leaching the entrapped DBR to create more voids and pores for water permeation.

## 1. Introduction

Membrane filtration has been recognized as one of the best methods for surface water purification and wastewater treatment [1]. It offers several advantages, including low energy consumption, simple operation at room temperature, low footprint, and can easily be combined with other processes [2]. Ultrafiltration is effective in separating dissolved macromolecules and small suspended particles from the feed solution [3,4].

Despite its advanced adoption in the industry, the feasibility of the membrane process can still be enhanced, most commonly via membrane material developments. Most of the polymeric membranes are produced in full-scale by using the phase inversion method [5,6]. The basic protocol is by dissolving a polymer into its solvent and cast into a thin film, followed by immersion in a coagulation bath containing its nonsolvent (mostly water). The protocol can be modified and optimized, including incorporating additive or polymers blending to yield the most optimum membrane properties.

Polyethersulfone (PES) is a widely used polymer for fabrication of ultrafiltration membranes. PES has high mechanical and hydrolytic stability, as well as thermal and chemical resistance. A phase-inverted PES-based membrane typically has an asymmetrical cross-section bulk structure. The characteristics and performance of the PES-based membrane are affected by composition (additive, concentration, and solvent), the temperature of the doping solution, solvent and nonsolvent affinity, coagulation bath, and environmental conditions [7]. Since most applied nonsolvent is water, the polymer for membrane fabrication, including PES, is typically hydrophobic. Without dosing additive or post-fabrication modifications, the resulting plain membranes are typically hydrophobic that susceptible to membrane fouling [8,9]. In addition, in some applications, a membrane with a large pore size is preferred to obtain high permeability, typically achieved by lowering the polymer concentration in the dope solution. However, due to low polymer matric packing density, membranes formed from low polymer concentration are highly porous with large macrovoids making them mechanically weak [10]. When used in the pressure-driven system, they suffer from severe compaction [11] even at low pressure [12]. Therefore, a method for membrane fabrication from low polymer concentration but having fewer macrovoids and high resistance from compaction is required [13,14].

Various approaches have been reported for membrane developments, mainly focusing on incorporating hydrophilic functional groups on the membrane surface to impose antifouling properties. There are three main methods, namely: (1) surface post-treatment such as coating and grafting; (2) blending of polymers during fabrication; and (3) incorporating additives during fabrication. The hydrophilic functional groups often added to PES-based membrane are generally derived from the sulfone, carboxyl, hydroxyl, and amine functions [13,15]. Recently, there have also been many reports of nanocomposite (combination of organic and inorganic). However, they are more costly and still contain hazardous materials [7].

Bio-based additives have recently come to the fore to assist in membrane developments because they are considered more sustainable and environmentally friendly. They include nano silica from rice husks and bagasse [16], nano carbons from palm oil shells [17], activated carbon from castor seeds [18], as well as ginger extract [19], and many others. These additives are low in molecular weight and pose hydrophilic properties due to their richness in polar groups. The incorporation of polar groups (i.e., hydroxyl (–OH), carboxyl (–COOH), amino (–NH_2_)) into the polymer matrix is considered to be able to increase the hydrophilicity of the membrane. This study explored a new type of bio-based resin polymer in form of dragonbloodin resin (DBR) for blending with PES to form ultrafiltration membranes.

DBR is secreted by red rattan. DBR grows in Indonesia, generally comes from the genus *Daemonorops* spp. It grows wild in some forests of Sumatra and Kalimantan but has been widely cultivated. This resin has a wide range of medicinal uses such as hemostatic, antidiarrheal, antimicrobial, antiviral, wound healing, anti-tumor, anti-inflammatory, and antioxidant [20]. There have been several reports on the antioxidant activity and potential of compounds isolated from the resin. The chemical components and structure of DBR can be seen in Figure 1. The DBR is rich in flavonoids, phenols, chalcones, and carboxylic acid [21], which contain hydroxyl groups. When DBR was introduced in the dope solution, some DBR could remain in the PES matric and introduced the hydroxyl and other polar groups in DBR to impose hydrophilicity.

The objective of this study was to investigate the effects of blending DBR into the dope solution on the characteristics and performances of the resulting PES-based membranes. Additive blending is one of the simplest methods and has been proved effective to alter the resulting membrane properties. It is expected that blending DBR and PES for membrane fabrication can enhance the bulk structure, alter the surface chemistry by introduction of polar chemical groups in DBR (Figure 1), and improve the filtration performance of the resulting membranes. DBR was added into the polymer solution to replace the PES partly (i.e., when 1% of DBR was added, 1% of PES was reduced). After preparation, the resulting membranes were characterized in terms of clean water permeability, water contact angle (WCA), surface chemistry using a Fourier transform infrared (FTIR) spectroscopy, surface and cross-section morphology using a scanning electron microscopy (SEM), also porosity, pore size, and water uptake. Later, the hydraulic performance in term of flux and rejection were conducted by filtration of humic acid solution. Lastly, the antifouling properties of the membrane samples were evaluated using the flux recovery ratio parameter.

## 2. Materials and Methods

### 2.1. Materials

PES (Ultrason E 6020 P, MW of 58 kDa, BASF, Germany) was used as the primary polymer, and 1-N-methyl-2-pirrolidone (NMP, 99.5%, Merck, Darmstadt, Germany) was used as the solvent. DBR powder as a bio-based polymer blend was purchased locally (Central Aceh, Indonesia). Humic acid (sodium salt, technical grade 50–60%, Sigma Aldrich, St. Louis, MO, USA) was used as an artificial feed solution for the rejection and the antifouling performance test.

### 2.2. Preparation and Characterization of the Dragonbloodin Resin

The DBR was in the form of a red resin powder. Upon purchasing, it was further refined using a mortar and pestle, followed by sieving using an 80-mesh sieve. FTIR spectroscopy (Shimadzu Prestige FT-IR 6400) was used to characterize the DBR powder to identify the presented chemical bonds. The FTIR spectrum was analyzed in the range 4000–400 cm^−1^ at room temperature.

### 2.3. Membrane Preparation

Four flat sheet membranes were prepared via the nonsolvent induced phase separation (NIPS) method. Table 1 detailed the composition of the dope solutions used for membrane preparation. Firstly, DBR powder was mixed into NMP for 20 min and stirred at a constant speed of 250 rpm. Subsequently, PES was introduced into the DBR/NMP mixture. The mixture was then stirred at 250 rpm for at least 24 h at room temperature to form a homogeneous dope solution. The solution was then left idle for overnight to remove any entrapped air bubbles. As depicted in Table 1, for particular addition of DBR, an equal amount of PES was reduced. Therefore, all dope solutions contained precisely the same number of solutes (PES + DBR). Preliminary study results showed that the addition of DBR to NMP up to 3% (*w*/*w*) could fully dissolve in NMP completely, but it was less soluble at concentration of above 3% (*w/w*).

Each dope solution was cast using a casting knife (YBA-3, BYK, Wesel, Germany) at 300 µm wet casting thickness on a glass plate. All membranes samples were prepared in one day to avoid variation in room temperature and humidity. After casting, the casted film was immersed immediately into a coagulation bath containing distilled water, acting as the nonsolvent in the NIPS. During the immersion process, a thin layer of the polymer film was formed. After a while (5–10 min), it floated from the glass plate indicating the completion of the NIPS. Subsequently, the formed flat sheet membrane was collected and washed with running tap water for 5–10 min to leach any residual NMP solvent. Finally, the membrane sheets were stored in a water container until further used for filtration tests and characterization.

### 2.4. Membrane Characterization

SEM device (Jeol, JSM-6360LA, Tokyo, Japan) was used to visualize the microstructure and morphology of both the top surface and the cross-section of the membranes. Before measurement, the samples were dried at room temperature. For the cross-section sample it was fractured under liquid nitrogen to obtain a clear cut. The SEM samples were coated with gold to provide conductivity.

According to a method detailed elsewhere, the overall porosity (ε, %) was determined from the SEM cross-section images that were processed with ImageJ software [22]. For each sample, the image was segmented into three, and each segment was processed with ImageJ. The porosity values obtained from the three segments were then presented as average with standard deviation. The membrane thickness (l) was also obtained from the cross-section SEM images. The Guerout–Elford–Ferry equation (in Equation (1)) was used to determine membrane mean pore radius $r_{m}$ (nm) based on the clean water permeability and the porosity data.
(1)$$r_{m}=\sqrt{\frac{\left(2.9-1.75\;\varepsilon \right)8\;ɳ\;l\;Q}{\varepsilon \;A\;\Delta P}}$$
where ɳ is the water viscosity (8.9 × 10^−4^ Pa·s), Q the volume of the permeate pure water per unit time (cm^3^/s), l membrane thickness (cm), A membrane surface area (cm^3^), and ΔP the operation pressure (Pa).

The WCA of the membrane was obtained by a high-resolution camera. Each membrane sample was tested at 8 points, and the results were averaged. The chemical bonds and functional groups near the membrane surface were identified using FTIR spectroscopy (Perkin Elmer Inc. Waltham, MA, USA). The IR spectrum was measured in a wavenumber range of 500–4000 cm^−1^. The chemical bonds and functional groups near the top of the membrane matric were identified based on the IR Spectrum table.

### 2.5. Membrane Filtration and Rejection Test

A dead-end filtration setup was used operated under a constant operating trans-membrane pressure (TMP or ΔP) of 1.5 bar and by mounting an effective membrane area of 11.34 cm^2^. After being mounted into the filtration cell, the membrane was first compacted by filtering clean water for one hour. Subsequently, the clean water flux was measured using filtration data of 1 h. The weight of the collected permeate was recorded every 10 min to obtain the evolution of the flux as a function of filtration time. The flux of pure water was then calculated using Equation (2). The same equation was used later to calculate the flux of humic acid solution filtration.
(2)$$J=\frac{V}{A\;t}$$
where J is the flux of pure water (L/(m^2^·h)), V the permeate volume (L), t the filtration time (h), and A the membrane surface area (m^2^).

The selectivity of the membrane was obtained by measuring the humic acid rejection from filtration of 50 mg/L humic acid solution. The humic acid concentration was analyzed using a UV-Vis Spectrometer (Shimadzu UV-1700, Kyoto, Japan). The humic acid rejection was calculated using Equation (3).
(3)$$R=\frac{C_{f}-C_{P}}{C_{f}}\times 100\%$$
where R is the rejection coefficient (%), C_f_ the humic acid concentration in the feed (mg/L), and C_p_ the humic acid concentration in the permeate (mg/L).

### 2.6. Antifouling Test

Flux recovery ratio (FRR) parameter was used to evaluate the membrane fouling propensity of the prepared membranes, which was obtained from a series of filtration stages. The first step was filtration of pure water for 60 min, followed by filtration of 50 ppm humic acid solution for 60 min. After that, the fouled membrane was physically cleaned. It was taken out from the filtration cell and washed by immersion in a beaker filled with distilled water for 20 min under magnetic stirring (at 50 RPM). The cleaned membrane was then reinstalled in the filtration cell and used for the filtration of distilled water. Those processes were considered as one full filtration cycle. The fouling filtration tests were conducted for five cycles for each membrane sample. The FRR was calculated using Equation (4). Membrane fouling behavior was also assessed through total fouling ratio (RT, %), reversible fouling ratio (RR, %), and irreversible fouling ratio (RIR, %). These three parameters were determined using Equations (5)–(7), respectively.
(4)$$\mathrm{FRR}=\frac{J_{Wi}}{J_{W\left(i+1\right)}}\times 100\%$$
(5)$$\mathrm{RT}=\frac{J_{Wi}-J_{HA}}{J_{wi}}\times 100\%$$
(6)$$\mathrm{RR}=\;\frac{J_{W\left(i+1\right)}-J_{HA}}{J_{wi}}\times 100\%$$
(7)$$\mathrm{RIR}=\frac{J_{Wi}-J_{w\left(i+1\right)}}{J_{wi}}\times 100\%$$
where $\;J_{W}\;$ and $J_{HA}$ were the water flux and humic acid flux, respectively. Meanwhile (*i*) and (*i* + 1) denote the water flux before filtration of humic acid in each filtration cycle.

### 2.7. Statistical Analysis

The results on membrane characterizations and assessment of filtration performances were analyzed statistically. The significance of a parameter was evaluated using a one-way analysis of variance (ANOVA) at 95% confidence intervals (*p* < 0.05). Then, the Tukey HSD post hoc was performed to identify which pairs of means were significantly different for multiple means comparison.

## 3. Results and Discussion

### 3.1. Chemical Compound of Dragonbloodin Resin

Figure 2 shows FTIR spectra that was obtained to identify the chemical bonds of the DBR. The C=C bending in the alkene bond could be identified at the peaks of 632, 700, 759, 823, 912, and 1010 cm^−1^. Peaks of 1602 and 1647 cm^−1^ indicated the stretching of the alkene. Then, the peaks in the range 1050–1310 cm^−1^ (1112, 1139, 1197, 1251, 1276 cm^−1^) were attributed to the C-O stretching in the carboxylic acid bonds. Peaks at 1350 and 1423 cm^−1^ fit well with spectra of O-H bending in phenol and a carboxylic acid, respectively, while peaks at 1730 and 1803 cm^−1^ indicated the C=O stretching. The wavenumbers range of 2840–3000 cm^−1^ for peaks of 2846, 2939, and 2999 cm^−1^ can be ascribed for the C–H group of alkenes. Meanwhile, a peak at 3062 cm^−1^ indicates O-H from the carboxylic acid.

The chemical bonds detected in the spectra suggest the three main elements of the DBR building block, which are C, H, and O. The data are consistent with an earlier report, in which DBR contained several components of dracorhodin, biflavonoids, polysaccharide, flavan, dicoflavan, abietic acid, and dammaradienol. All have the chemical formula C_x_H_y_O_x_ [20]. Similar results were also reported by [23,24], with DBRs obtained from different species. Following this result, DBR as a novel bio-additive in membrane matric can provide carboxyl and phenol groups, both of which are hydrophilic [13], which could enhance the wetting property of the DBR loaded membranes.

Figure 2 also shows that the peaks of DBR resemble the peaks of PES, particularly in the wavenumber range of 500–1700 cm^−1^. It was hypothesized that apart from containing hydrophilic functional groups, DBR building block consists of a long chain of saturated hydrocarbon as the backbones, which reduces the overall polarity of the compounds. Therefore, it is expected that a fraction DBR would remain in the polymer matric, as demonstrated by the color of the membrane in the insets of Figure 1.

### 3.2. Membrane Characterization

#### 3.2.1. Surface Water Contact Angle

Figure 3 shows the WCA and water uptake of the prepared membrane showing that higher loadings of DBR led to higher wettability of the resulting membranes. The WCA of M-0, M-1, M-2, and M-3 were 74.0° ± 0.8°, 66.7° ± 2.4°, 55.3° ± 0.5°, and 52.0° ± 0.8°, respectively. The WCA was measured on the feed-side surface or the top side of the membrane sheet on the glass plate during the NIPS, which quantitatively identifies its hydrophilicity. WCA is affected by surface chemistry from the presence of polar or non-polar functional groups and the surface structure or topography [25,26].

The findings in Figure 3A suggest that DBR loading lowered the WCA for most of the pairs. Statistical analysis results using the Tukey HSD suggested that loading of DBR up to 2 wt% significantly increased the membrane surface wettability (*p*-values < 0.001), and the increment was not significant for loadings of 2 and 3 wt% (with *p*-value = 0.134). The increase in membrane surface wettability can be explained by the presence of polar groups of the DBR (Figure 1). As demonstrated later, the higher wettability of the membranes loaded with DBR was expected to impose the antifouling property. Apart from the polar functional groups, WCA is also well known to be affected by the surface morphology and topography [27].

The results of WCA were consistent with the trend of the water uptake, in which the membrane with low WCA posed a higher water uptake (Figure 3B). Membrane with good wettability embedded more water thanks to its hydrophilic property. Membrane with hydrophilic property has a strong interaction with water and could form a hydration layer that reduces the interaction with foulant materials [28]. The higher water uptake could also be caused by change in the pore structure, which was related to the bulk porosity. The higher DBR loading led to formation of more porous membrane. Similar to the WCA, statistical analysis results using the Tukey HSD suggest that loading of DBR up to 2 wt% significantly increased the membrane water uptake (*p*-values < 0.001), and the increment was not significant for loadings of 2 and 3 wt% (*p*-value = 0.216).

#### 3.2.2. FTIR Spectra of the Membrane Samples

The FTIR spectra of all membrane samples indicated no difference in the appearance of peaks (Figure 1). A peak at a wavenumber of 1577 cm^−1^ typically identifies the characteristics of the PES membrane functional group representing the C=C stretching in cyclic alkanes. Peaks at 1317, 1297, 1238, and 1148, 1577 cm^−1^ corresponded to sulfone group groups (S=O stretching). In contrast, peaks at wavenumbers 1104 and 1071 cm^−1^ represented C–O stretching bonds. Finally, peaks at 1010, 871, 835, 717, 717, 700, and 627 cm^−1^ represented the C=C group stretching the alkanes [29,30,31].

The appearance of a unique peak associated with the DBR which is not clearly visible in M-1, M-2, M-3 because there are similar peaks between DBR and PES. However, the presence of DBR on the membrane was identified with a slight difference in intensity at several peaks in Figure 2. This difference was quite visible for membranes with 3% DBR (M-3) at peaks at 700, 823, and 1010 cm^−1^ represented C=C group of alkene stretching, and peaks at 1112 cm^−1^ represented C–O stretching bonds from the DBR. The higher wettability of the membranes containing DBR could then be attributed partly to the presence of polar groups on the membrane surface.

#### 3.2.3. Morphology

Based on the surface SEM images, all membranes generally had similar flattened surface morphology (Figure 4). Overall cross-section images demonstrate the typical structure of phase-inverted membrane comprising of dense top layer atop a porous supporting layer. However, a clear trend can be seen from the cross-section SEM images in which loading DBR additive increased the bulk porosity. The overall structure changed from the macrovoid-rich supporting layer in M-0 and M-1 into finger-like voids for M-2 and M-3. The combined effect of kinetic (viscosity) and thermodynamic (demixing) could explain the overall membrane structure. Membranes made from low polymer concentrations resulted in more porous structure with large pore size [32,33]. The finding opens a new avenue for the preparation method of ultrafiltration membrane via phase inversion by incorporating nature-based polymer (i.e., DBR). Partial substitution of PES by bio-based polymer resulted in a similar bulk structure of membrane without DBR biopolymer, which allowed membrane preparation using low polymer concentrations to form a membrane with a large pore size. Such membrane is expected to offer substantially high water flux, as discussed in Section 3.3.

#### 3.2.4. Pore Size, Porosity and Pure Water Flux

Figure 5 shows that loading of DBR increased the membrane pore size. Small loading of 1% DBR in M-1 resulted in an increase of pore size by a factor of two compared to the neat PES (M-0). Loading of 2–3% DBR increased the pore size fourfold from M-0. Such high increment is reflected from drastic change of the bulk structure from macrovoids to finger-like morphologies, as shown in Figure 4.

Figure 5B shows that the bulk porosities of the membrane samples containing DBR were generally higher than M-0, except for M-1. The porosities were very high of >75%, a typical phase inverted membrane produced from instantaneous demixing. However, the variability of the porosity between M1, M2, and M3 was not very high. The results agree with the cross-section images. The porosity appeared to be similar, and the void’s shapes changed from macrovoids (M-0 and M-1) to finger-like (M2 and M4).

The rate of demixing can explain the trend in increasing pore size during the NIPS. A highly porous membrane with large macrovoids and large pore size is associated with fast or instantaneous demixing from poorly stable dope solution [34]. The rate of demixing and the interaction between polymers, solvents, and nonsolvent can be explained using the Hansen solubility parameter or cloud point experiment. In this context, the cloud point test was untenable because of the red color of the dope solutions (Figure 2). In addition, Hansen’s solubility parameters and interaction of DBR with the solvent could not be predicted because of the unknown chemical structure of DBR. Only limited information is available on DBR identification, including the Raman spectra and FTIR spectra [35]. Loading of DBR somewhat affected the path of phase inversion in which higher loading leads to the formation of larger pore size (more instantaneous demixing). The hydrophilic functional groups in DBR make it acting as nonsolvent and destabilized the polymer solution. Consequently, more instantaneous demixing was expected, leading to a more porous membrane with larger—and possibly more—pores.

The pure water fluxes of M-0, M-1, M-2 and M-3 were 9.89, 36.78, 231.41 and 246.79 L/(m^2^·h), respectively (Figure 5C). Loading of 3% DBR into the dope solution increased the pure water flux by 25-folds. Pristine PES (M-0) has a very low clean water flux of 9.89 L/(m^2^·h), most likely because of its small pore size (0.0103 µm) and dense membrane surface morphology. The increase in clean water flux was 3.7-folds when comparing M-1 to M-0, most likely to both an increase in average pore size to 0.0218 µm and an increase in the number of pores in M-1. Membranes can have exactly similar pore size but differ in the number of the pore, as detailed elsewhere [36]. Similar justification can be attributed to the clean water fluxes of M-2 and M3, with 24 and 25-folds higher than M-0, respectively. M-2 and M-3 had pore sizes of 0.0418 µm and 0.0418 µm, respectively. Based on the analysis of the results, all of membranes were in ultrafiltration range.

### 3.3. Hydraulic Performance

#### 3.3.1. Humic Acid Filtration

The results of the membrane characterization supported the finding on the humic acid solution filtration, in which M-3 showed the highest flux of 154.5 ± 6.6 L/(m^2^·h), 30-folds higher than M-0 without substantial loss in rejection (Figure 6). However, humic acid fouling led to flux loss by 38% compared with the pure water flux and 60.4 ± 0.9% rejection of humic acid. M-0, M-1 and M-2 had fluxes of 5.2 ± 0.9, 32.7 ± 2.5, and 145.5 ± 6.6 L/(m^2^·h), while rejections of 71.1 ± 2.1, 70.5 ± 3.0, and 58.0 ± 0.9%, respectively. The humic acid rejection decreased along with an increasing concentration of additives due to larger pore sizes that allowed permeation of humic acid.

Similar to the pure water flux, the humic acid flux and rejection trends can be explained by the larger size and number of pores for the DBR loaded membranes (M-1, M-2, M-3). Other parameters that determine the separation of solutes are surface changes and hydrophilicity [37]. Referring to previous research [16], the humic acid used in this work had a molecular weight of 226 Da corresponding to a minimum diameter of 0.8 nm, which was smaller than the estimated pore size hence could not fully rejected. The filtration of humic acid occurred both on pore and in pore of membrane. The selection of humic acid in this study was based on its abundance in surface water along with other compound generally termed as natural organic matters.

#### 3.3.2. Antifouling Test

The effect of fouling on the membrane is represented by the loss of flux over the filtration time (Figure 7). Each membrane showed a decrease in flux over filtration time due to the occurrence of membrane fouling. Substantially lower fluxes demonstrated the occurrence of membrane fouling in comparison to the clean water fluxes. Fortunately, a simple water flushing restored the flux-loss effectively. Compared to the DBR loaded membranes, the flux evolution of M-0 was relatively flat, indicating very low fouling, but extremely low in water flux. When considering the filtration throughput and the restorability of permeation, dosing DBR to the PES-based membranes offers a substantial advantage in terms of hydraulic throughput. M-0 offered prolonged filtration operation with a low fouling rate but would require a substantially large membrane area to achieve the same throughput in comparison to other DBR loaded membranes. The flux decline on the DBR loaded membranes can quickly be restored via simple water washing.

Data from the five filtration cycles shown in Figure 7 was further analyzed to investigate the flux reduction profile to determine the antifouling parameters for each membrane sample. The FRR represents the ability of the membrane to recover its performance after being fouled, thereby minimizing flux reduction after successive filtration. Five filtration cycles were carried, and the results are shown in Figure 8. The FRRs of each cycle for M-0 were >80%, demonstrating a very high recoverability of fouling. However, it should be noted that the flux was very low compared to the DBR loaded membranes.

The fouling parameters for M-1 were quite close to M-0, which agrees with both membranes’ characteristics as detailed earlier. While for M-2 and M-3, the FRRs were >90%, even after several cycles. In some cases, the FRRs were >100%, with the RIR of <0%. These findings indicated that both membranes could restore fouling and gain additional flux after cleaning (water flushing). This trend can be seen in Figure 7, where the initial and final flux fluxes after each cycle slightly increased as the filtration cycle progressed. The pattern resembles the phenomenon of additive leaching in an earlier report [38]. The membrane pore size enlarged as the filtration progressed. We suspect the same phenomena in this study, in which the residual DBR in the PES matrix leached out during the filtration or the cleaning. Nonetheless, detailed mechanisms of DBR leaching need to be investigated as a follow-up study.

## 4. Conclusions

The PES/DBR ultrafiltration membrane was successfully prepared and showed good characteristics and performance. The hydrophilicity, porosity, morphology, and water uptake increased along with the increase in DBR loading. The DBR loaded membranes tended to have more and larger size of pores. Those properties were translated into higher permeabilities. The PES-based membrane loaded with DBR reached the clean water flux of 246.79 L/(m^2^·h), 25-folds higher than the pristine PES membrane. The humic acid solution flux reached 154.5 ± 6.6 L/(m^2^·h), 30-folds higher than the pristine PES membrane without substantial loss in rejection (71 vs. 60%). All DBR loaded membranes also showed excellent antifouling properties as shown by the FRR values of almost 100%. The hydraulic performance could still be enhanced by leaching out some entrapped DBR to create voids on more pores, which will be the subject of the follow-up study.

## Figures and Tables

**Figure 1 polymers-13-04436-f001:**
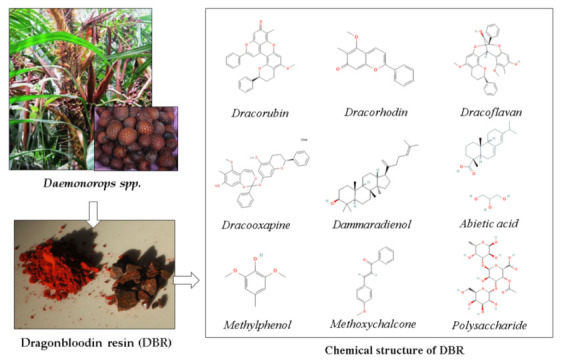
Chemical component of DBR (adapted from [20]).

**Figure 2 polymers-13-04436-f002:**
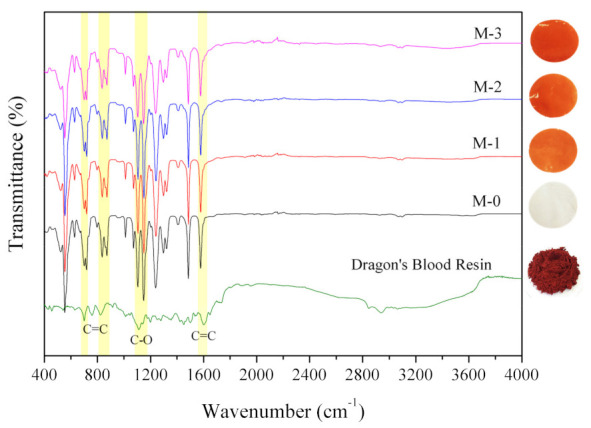
FTIR spectra of the DBR powder and the membrane samples. The insets show the appearance of the membrane samples resembling the uniformly distributed DBR loadings.

**Figure 3 polymers-13-04436-f003:**
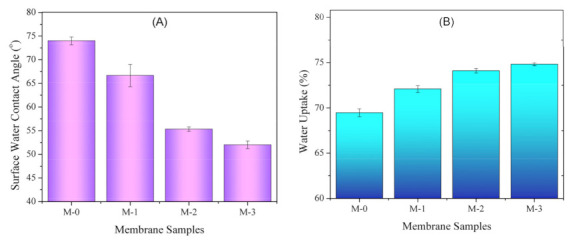
Surface water contact angle (**A**) and water uptake rate (**B**) of the membrane samples.

**Figure 4 polymers-13-04436-f004:**
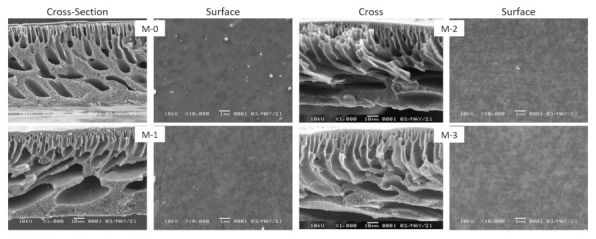
Cross-section and surface SEM images of the membrane samples.

**Figure 5 polymers-13-04436-f005:**
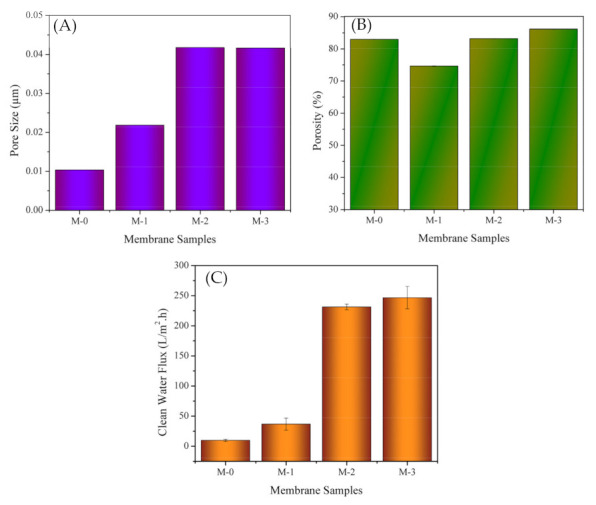
The pore size of the membrane samples estimated using the Guerout–Elford–Ferry equa-tion (**A**), membrane porosity using ImageJ (**B**), and the clean water flux (**C**).

**Figure 6 polymers-13-04436-f006:**
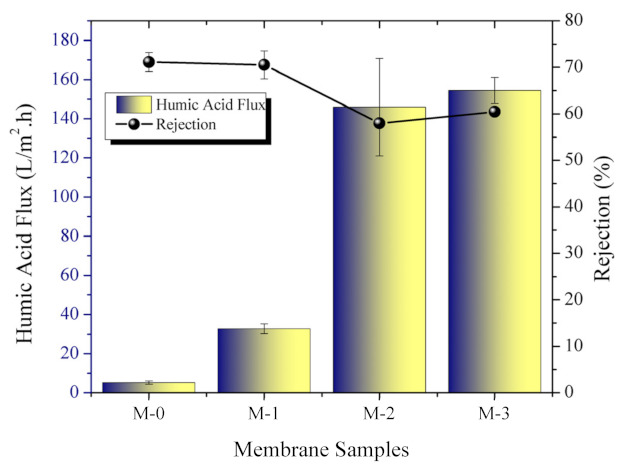
Humic acid flux and rejection of each membrane sample.

**Figure 7 polymers-13-04436-f007:**
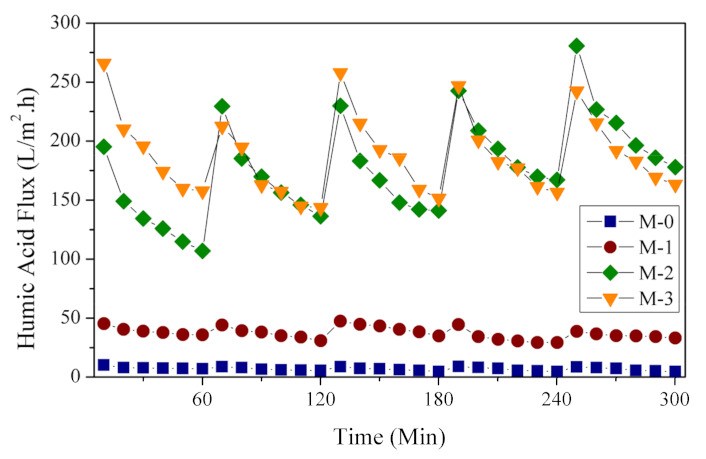
Reduction and recovery of humic acid flux during five filtration cycles.

**Figure 8 polymers-13-04436-f008:**
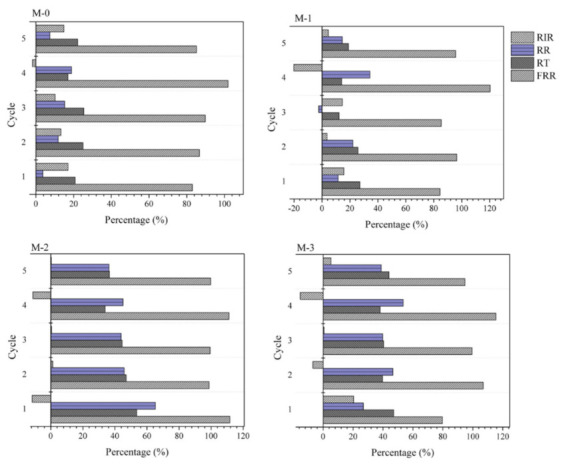
Antifouling parameter of each membrane.

**Table 1 polymers-13-04436-t001:** Composition of the prepared PES membranes.

Membrane Code	PES (% *w*/*w*)	DBR (% *w*/*w*)	NMP (% *w*/*w*)	DBR/PES(%)
M-0	17.5	0	82.5	0
M-1	16.5	1		1/16.5
M-2	15.5	2		2/15.5
M-3	14.5	3		3/14

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
