# Peer review of "Improvement of Properties and Performances of Polyethersulfone Ultrafiltration Membrane by Blending with Bio-Based Dragonbloodin Resin"

_polymers, 2021, doi:10.3390/polym13244436_

Round 1
Reviewer 1 Report
The manuscript includes the membrane preparation with blend of polyethersulfone and some resin. The manuscript concept is acceptable and its structure is of readable. some points should be revised and added, against the following comments,
1) structure of dragonboodin resin is not cleared. what kinds of molecules are existed, and what kinds of hydrophobic group are there?
2) in introduction mention the interaction of PES and added resin, Was is only entanglement, or some specific interaction takes place?
3) what is the molecular weight of PES?
4) in adding DBR to NMP was is completely dissolve?
5) in Table 1 molar ratio of PES and DBR is better to add, then reader can imagine the blending situation of polymers.
6) in IR, is there any shift of peak with blending?
7) in thickness of membrane, DBR is uniformly distributed?
8) on page 10, line 3, filtration of humic acid occurred on pore or in pore of membrane?
9) fouling phenomena is dependent on applied pressure in permeation. can check them?
Reviewer 2 Report
In the work “Improvement of properties and performances of polyethersulfone ultrafiltration membrane by blending with bio-based dragonbloodin resin”, the authors prepared a series of PES UF membranes with different PES to DBR ratio, while keeping the total solid concentration constant in the casting solution. The membranes with higher DBR loading generally gave better filtration performance, e.g., higher flux and lower degree of fouling, than the pristine PES membranes, owning to the improved hydrophilicity and larger pore size provided by the DBR. Overall, it seems a solid piece of work with some practical application value. However, I think some experimental results could be better explained and the novelty of this work should be further justified. Therefore, I would recommend the paper to be published after major revision. Below are my comments that I hope the authors could consider/address properly.
- There are many similar UF membrane modification works published in the past few years. What is the major reason of using DBR? Is DBR-based modification more economic/environmentally friendly than the modifications based on other molecules? Please add more justification to address this point.
- Page 1, line 38. Does the “low footprint” mean “low carbon footprint”?
- Page 3, line 104. The “Humid acid” here should be “Humic acid”.
- Page 3, line 125-127. Was the room temperature and humidity controlled during the casting process?
- Page 4, line 178-180. The physical cleaning method described here seems to be gentle. Could it provide sufficient cleaning?
- Page 6, line 242-244. While the DBR modification indeed improved the membrane hydrophilicity, it also changed the pore structure. I was wondering if the change in pore structure also contributed to higher W.U.?
- Page 7, line 261-265. To make the difference in intensity more visible, I suggest the authors normalize all lines in the FT-IR spectra.
- Page 9, Figure 5. Please re-plot this figure as the columns for M-0 and M-3 did not show properly in the current figure.
- Page 11, line 379-381. The authors attribute the phenomena of flux increase to the leaching of DBR from the PES matrix during the filtration process or the cleaning process. This observation brings up several questions. First, the clean water flux obtained in this work might not be accurate enough for all DBR containing samples. Second, the “true” anti-fouling effects from DBR could not be fully revealed. I suggest the authors perform sufficient hydraulic cleaning on all DBR modified membranes, i.e., till the pure water flux and the water contact angle reach constant values, before running HA filtration tests.
Round 2
Reviewer 1 Report
The manuscript is well revised against the reviewer's comments, but still introduction is better to be revised. the comments are as follows,
1) on page 2, line 74-81. more concretely mention the chemical structure's information of bio-based additive to the membrane. with blending of bio additives, what is enhanced and explain more chemically.
2) line 91, entanglement should be changed to unclear.
3) line 96-, more chemical mention the blending advantage and strategy of DBR and PES. Why pore and hydrophilicity is enhanced what is the role of phenol group, and -OH?
Author Response
Reviewer 1
The manuscript is well revised against the reviewer's comments, but still introduction is better to be revised. the comments are as follows,
- on page 2, line 74-81. more concretely mention the chemical structure's information of bio-based additive to the membrane. with blending of bio additives, what is enhanced and explain more chemically.
The hydrophilicity of membrane can be enhanced by blending bio additives, with incorporating the polar groups (i.e, hydroxyl (-OH), carboxyl (-COOH), amino (-NH2)) into the polymer matrix. See revised manuscript Lines 79-81.
- line 91, entanglement should be changed to unclear.
The interaction between PES and DBR was entanglement by introducing the hydroxyl groups to the backbone polymer chain. See revised manuscript Lines 91-93.
- line 96-, more chemical mention the blending advantage and strategy of DBR and PES. Why pore and hydrophilicity is enhanced what is the role of phenol group, and -OH?
Thank you for the suggestion. We have applied some changes in response to the suggestion. See revised manuscript Lines 99-102.
Reviewer 2 Report
The authors have addressed the comments from the reviewers properly, and the quality of the paper has been improved. I'm pleased to recommend the paper for publication in its current form.
Author Response
Thank you for the encouraging comment.